# Temporal and Spatial Dynamics of Tumor–Host Microbiota in Breast Cancer Progression

**DOI:** 10.3390/microorganisms13071632

**Published:** 2025-07-10

**Authors:** Qi Xu, Aikun Fu, Nan Wang, Zhizhen Zhang

**Affiliations:** 1Ocean College, Zhejiang University, Zhoushan 316021, China; 3160102619@zju.edu.cn (Q.X.); n_wang@zju.edu.cn (N.W.); 2Sichuan Clinical Research Center for Medical Imaging, Dazhou 635000, China

**Keywords:** breast tumor, bacteria, splenomegaly, gut microbiota, spatiotemporal trajectory progression

## Abstract

Deciphering the spatiotemporal distribution of bacteria during breast cancer progression may provide critical insights for developing bacterial-based therapeutic strategies. Using a murine breast cancer model, we longitudinally profiled the microbiota in breast tumor tissue, mammary gland, spleen, and cecal contents at 3-, 5-, and 7- weeks post-tumor implantation through 16S rRNA gene sequencing. Breast tumor progression was associated with lung metastasis and splenomegaly, accompanied by distinct tissue-specific microbial dynamics. While alpha diversity remained stable in tumors, mammary tissue, and cecal contents, it significantly increased in the spleen (*p* < 0.05). Longitudinal analysis revealed a progressive rise in Firmicutes and a decline in Proteobacteria abundance within tumors, mammary tissue, and cecum, whereas the spleen microbiota displayed unique phylum-level compositional shifts. Tissue- and time-dependent microbial signatures were identified at phylum, genus, and species levels during breast tumor progression. Strikingly, the spleen microbiota integrated nearly all genera enriched in other sites, suggesting its potential role as a microbial reservoir. Gut-associated genera (*Lactobacillus*, *Desulfovibrio*, *Helicobacter*) colonized both cecal contents and the spleen, with *Lactobacillus* consistently detected across all tissues, suggesting microbial translocation. The spleen exhibited uniquely elevated diversity and compositional shifts, potentially driving splenomegaly. These results delineated the trajectory of microbiota translocation and colonization, and demonstrated tissue-specific microbial redistribution during breast tumorigenesis, offering valuable implications for advancing microbiome-targeted cancer therapies.

## 1. Introduction

Breast cancer accounts for a quarter of all cancers and is the second dominant factor of cancer-related deaths worldwide, which is a severe threat to the physical and psychological health of women [1,2,3]. It has been demonstrated that the microbiota in the tumor and gut play pivotal roles in breast health and carcinogenesis [4]. Extensive studies have been conducted to understand the relationship between breast microbiota, gut microbiota, and breast tumors. Alterations in microbial diversity in breast tumors and the gut were associated with breast carcinogenesis by modulating host immune responses and inflammatory pathways [5].

It was established that the microbial spectrum in breast tumors was clearly different from healthy breast tissue [6]. The predominant microbial community in normal breast tissue includes phyla of *Proteobacteria*, *Firmicutes*, *Actinobacteria*, and *Bacteroidetes* [7,8], which were abundant in *Sphingomonas*, *Lactobacillaceae*, *Acetobacterraceae*, *Leuconostocaceae Xanthomonadaceae*, *Cyanobacteria*, and *Prevotellaceae*, whereas these bacteria were inadequate in breast tumors [9]. These normal breast microbiota produced useful biomaterials to keep a healthy micro-environment of the breast tissue [10] and were associated with beneficial functions such as producing short-chain fatty acids (SCFAs), promoting immunological responses, inducing fructose and mannose metabolism, increasing cysteine and methionine metabolism, glycosyltransferases biosynthesis, degrading environmental carcinogens [11,12,13,14]. Compared to normal breast tissues, the abundance of microbial communities in breast tumors was significantly different. Breast tumor microbiota were abundant at the genus level in *Fusobacterium*, *Atopobium*, *Gluconacterobacter*, and *Hydrogenophaga*. These bacteria were associated with dysregulation of cell proliferation, metabolic pathways, and immunological responses, contributing to tumor growth and progression [6,15,16]. *Enterobacteriaceae* and *Staphylococcus* that enriched in breast tumor [17], can induce DNA double-strand breaks to result in malignant progression of breast cell through producing genotoxins [18,19], *Lactobacilli*, abundant in breast tumor microbiota could lead to chemotherapy and radiation resistance of tumors by producing lactic acid to lower pH ratio [20]. These bacterial taxa of *Enterobacteriaceae*, *Staphylococcus,* and *Lactobacilli* were also found to promote breast tumor metastasis and colonization [21].

Multiple studies have shown that gut microbiota also changes during breast tumor progression. The most dominant phyla of the gut microbiota are *Firmicutes* and *Bacteroidetes*, followed by *Actinobacteria*, *Proteobacteria*, *Fusobacteria,* and *Verrucomicrobia* phyla [22,23]. Consistent with the breast microbiota, there were distinct gut microbiota between breast tumor patients and healthy individuals. Previous studies revealed that *Bacteroidetes* and *Firmicutes* were significantly enriched in the gut of breast cancer patients compared with controls [24,25]. The gut microbiota affected the progression of breast cancer through the production of bacterial metabolites, the alteration of estrogen levels, and the modulation of immune system activity [26,27].

Recent evidence showed that there was a relationship between tumoral microbiota and gut microbiota. Gut microbiota could be transported to the breast tumor site through hematogenous spread to colonize in the tumor as intratumoral microbiota [28]. However, not all tumoral bacteria originated in the intestine; the tumor microbiota may be an intrinsic tumor component [21,29]. The translocation of gut bacteria to extra-intestinal tissues such as mesenteric lymph nodes (MLN) and spleen was accompanied by disease conditions that impair the intestinal epithelial tight junctions, inducing luminal bacteria to pass through the epithelial barrier into the bloodstream [30,31,32]. A recent study showed the translocation of gut microbiota to the tissues of MLN and spleen augmented extraintestinal anticancer immune responses [33], indicating spleen played a crucial role in tumor growth and therapy.

Although numerous studies have reported characteristic alterations in the gut and tumor microbiota of tumor-bearing mice, the compositional dynamics of the breast cancer microbiome remain poorly characterized. Furthermore, the impact of tumor progression on microbiota composition in the intestinal tract and extraintestinal organs, especially the lymphoid organ spleen tissue, has yet to be elucidated. Additionally, we observed that tumor-bearing mice frequently exhibit splenomegaly. Microbiological culture results showed the presence of bacteria in the enlarged spleen. The migration of gut bacteria to the spleen tissue with breast tumor growth is unclear. In this study, we employed a murine breast cancer model to systematically investigate and map the temporal variations in microbial composition within breast tumors, normal mammary tissue, spleen, and cecal contents throughout tumor progression. These findings delineate the spatiotemporal distribution of bacteria at various stages of breast cancer to assess the role of microbial translocation in shaping tumor microenvironments and the mechanism by which microbial–host interactions affect the development and progression of breast cancer. These results may provide fundamental insights into microbiota-targeted therapeutic strategies.

## 2. Materials and Methods

### 2.1. Mice Management

Female mouse mammary tumor virus-polyoma middle tumor-antigen (MMTV-PyMT) transgenic mice (FVB/N-Tg(MMTV-PyVT)634Mul/J, 002374), which develop spontaneous breast tumors, were obtained from Shang Chai Lab of Westlake University. Twenty-four female FVB mice, 6 weeks old, with an average initial weight of (25.3 ± 2.1) g, were purchased from the Experimental Animals Center of Zhejiang Province (Hangzhou, China). All animals were bred in the Experimental Animals Center of Zhejiang University, housed in specific pathogen-free conditions, and given standard mouse chow. All animals were kept on a 12-h light/12-h dark cycle and at room temperature for one week to adapt to the environment before the experiment, then subjected to the breast tumor growth experiment. The experiment period was 7 weeks. All animal experiments were carried out in strict accordance with the Guide for the Care and Use of Laboratory Animals at Zhejiang University. Experimental protocols for animal studies were approved by the Institutional Animal Care and Use Committee of Zhejiang University (No. ZJU20250286).

### 2.2. Murine Model of Breast Tumor

A breast tumor from the MMTV-PyMT model was screened. Dissociated spontaneous PyMT tumor cells were implanted into the unilateral mammary gland of female FVB mice at 1 × 10^5^ cells/mouse, and the contralateral healthy mammary gland from the same individual without implanting tumor cells was set as the control. Tumor growth in mice was monitored twice a week, and the mice were euthanized when tumors reached 2 cm in diameter as humane endpoints.

### 2.3. Breast Tumor Growth and Lung Metastasis Assay

In this study, blinding was applied during sample collection at each time point. At 3 weeks (3 w/Tw), 5 weeks (5 w/Fw), and 7 weeks (7 w/Sw) after implantation of the tumor cell, eight female FVB mice were euthanized at each time point. Tissue samples of the breast tumor and spleen were aseptically collected and weighed to calculate tumor growth and splenomegaly. Lung metastasis was detected to calculate the lung metastasis rate (*n* = 8).

### 2.4. The Tissues and Cecal Contents DNA Extraction

At 3 weeks (3 w/Tw), 5 weeks (5 w/Fw), and 7 weeks (7 w/Sw) after implantation of tumor cell, tissue samples of the breast tumor tissue, contralateral healthy breast tissue, spleen tissue and cecal contents of five female FVB mice were isolated for microbiome analysis (*n* = 5) at each time point. The DNA of the total bacterial community was extracted using E.Z.N.A.^®^ Bacterial DNA Kit (Omega Bio-Tek, Norcross, GA, USA) according to the manufacturer’s instructions. The quality of extracted DNA samples was measured using a NanoDrop spectrophotometer (Thermo Fisher Scientific, Waltham, MA, USA) and further confirmed by PCR amplification of the bacterial 16S rRNA gene.

### 2.5. Illumina High-Throughput Sequencing

The V3-V4 region of the 16S rRNA genes was amplified from the bacterial DNA template. The primers of 338F (5′-ACTCCTACGGGAGGCAGCA-3′) and 806R (5′-GGACTACHVGGGTWTCTAAT-3′) were used, which were modified with 7 bp barcodes to distinguish the PCR products. PCR products were detected by agarose gel electrophoresis and subsequently purified using an AxyPrep™ DNA gel extraction kit (Axygen, Hangzhou, China). Each purified PCR product was sequenced by Illumina high-throughput sequencing on the IonS5^TM^XL platform (Beijing E Hanbo Biotechnology Co., Ltd., Beijing, China).

### 2.6. Processing of Illumina Sequencing Data

Raw FASTQ file reads were quality-filtered using Cutadapt (V1.9.1) to remove low-quality part reads and chimeras to get the Clean Reads. Operational taxonomic units (OTUs) were defined as sequences clustered with a threshold of 97% Identity using Uparse software (V7.0.1001). Species annotation analysis was performed using the Mothur method and SSUrRNA database of SILVA132 (set the threshold as 0.8 to 1), and taxonomic information was statistically analyzed at taxa levels: kingdom, phylum, class, order, family, genus, and species. Normalized processing was carried out based on the standard of the least amount of data in the samples. The relative abundance and alpha/beta diversity analysis were conducted based on the normalized OTUs. Alpha diversity indices, including Observed-species, Chao1, Shannon, Simpson, ACE, Goods-coverage, and Beta diversity indice-UniFrac distance metrics were calculated using Qiime software (Version 1.9.1) and were used for assessing community diversity. The rarefaction curve and NMDS were plotted using R software (Version 2.15.3). The rarefaction curve evaluated the sequencing depth of each sample. The difference analysis of the Alpha diversity index and beta diversity index was conducted parametric tests, with Tukey’s test and non-parametric tests with the Wilcox test, respectively. The taxonomy compositions and abundances were visualized using GraPhlAn (Version 1.1.3). A Venn diagram was generated to represent the number of shared and unique species of OTUs among tissues. LDA Effect Size (LEfSe) analysis using LEfSe software (http://galaxy.biobakery.org/, accessed on 1 July 2025) analyzed species abundance between groups and screened out significantly different bacterial taxa between groups (LDA Score > 4). Microbial functions were predicted by PICRUST (version 1.1.0) based on the OTU tree and OTU genetic information in the Greengene database. Predicted functional pathways were annotated by using the Kyoto Encyclopedia of Genes and Genomes (KEGG) at level 2.

### 2.7. Data Analysis and Statistics

All data were presented as the Mean ± standard deviation (Mean ± S.D.) and were analyzed with Tukey’s post hoc test, Wilcox test, or unpaired *t*-test, as defined in the figure legends. * represents *p* < 0.05 for statistical significance and ** represents *p* < 0.01 for high significance. All the statistics were performed with SPSS 25.0 software (SPSS Inc., Chicago, IL, USA).

## 3. Results

### 3.1. Splenomegaly and Lung Metastasis with Breast Tumor Growth

Longitudinal monitoring revealed progressive tumor development and systemic manifestations in the breast cancer model. Quantitative analysis demonstrated statistically significant increases in both breast tumor mass (*p* < 0.05) and spleen weight (*p* < 0.05) at 5 weeks ((3.59 ± 1.35) g and (0.39 ± 0.09) g) and 7 weeks ((11.33 ± 3.32) g and (0.33 ± 0.18) g) post-implantation compared to baseline measurements at 3 weeks ((0.64 ± 0.56) g and (0.16 ± 0.05) g) (Figure 1a,b). Concurrently, we observed a time-dependent escalation in metastatic dissemination, with lung metastasis rates rising from 0% at 3 weeks to 50% (4/8 animals) by 5 weeks, ultimately reaching 87.5% (7/8 animals) at the 7-week endpoint (Figure 1c). These findings collectively demonstrate that tumor progression in this model is accompanied by clinically relevant systemic effects such as significant splenomegaly, suggesting potential immune system involvement or systemic inflammatory responses, and increasingly prevalent pulmonary metastases.

### 3.2. The Microbiota Richness and Diversity Analysis with Breast Tumor Progression

The high microbial community coverage (Goods coverage range from 0.986 to 0.997) indicated the eligible sequencing quality across all samples (Appendix A). The average read length was 409 bp. A total of 29,069 operational taxonomic units (OTUs) were identified across the sample matrix. Comparative analysis using Venn diagrams demonstrated a significant expansion of unique OTUs in two critical tissue compartments: breast tumors at 5 weeks and spleen tissues at 7 weeks (Figure 2a,b). We employed multiple established metrics to evaluate bacterial alpha diversity: Shannon and Simpson indices for species diversity assessment, and Chao1, ACE, and Observed species indices for species richness quantification [34,35]. This comprehensive approach allowed for robust characterization of both diversity and richness parameters within the microbial communities under investigation.

Alpha diversity analyses revealed tissue-specific patterns during tumor progression. The rarefaction curves of all samples indicated that the sequencing data volume was reasonable (Appendix A). While breast tumors, normal breast tissue, and cecal contents maintained stable alpha diversity metrics across time points (*p* > 0.05) (Figure 3a,b,d), spleen tissues exhibited significant temporal alterations in microbial richness. Specifically, spleen samples at 7 weeks demonstrated substantial increases in Chao1 (1048.88 ± 88.10, *p* < 0.05) and ACE (1033.82 ± 70.36, *p* < 0.05) indices compared to 5-week samples (Chao 1, 575.11 ± 210.90, ACE, 593.92 ± 226.17). Moreover, the Observed species estimator at 7 weeks (900.00 ± 39.09) was markedly elevated relative to both 3-week (532.80 ± 229.14) and 5-week (430.00 ± 169.14) time points (*p* < 0.01, Figure 3c). Cross-tissue comparisons at specific tumor stages identified distinct microbial profiles (Figure 3e). At 5 w, Chao1 and ACE of normal breast tissue (589.39 ± 103.94 and 570.94 ± 90.84) were significantly higher (*p* < 0.05) and highly significantly higher (*p* < 0.01) than those of cecal contents (329.92 ± 76.04 and 324.55 ± 63.16), respectively. The Shannon and Simpson values of the breast tumor (6.30 ± 1.72 and 0.89 ± 0.16) were significantly higher than those of spleen tissue (3.59 ± 0.77 and 0.75 ± 0.10) (*p* < 0.05), and the Observed species of breast tumor (888.20 ± 359.91) was highly significantly higher than that of cecal contents (430.0 ± 169.10) (*p* < 0.01, Figure 3e).

Beta diversity analysis using the unweighted UniFrac distance and unweithted count revealed distinct microbial community patterns across tissue types. The unweighted UniFrac distance demonstrated no significant temporal separation in microbial communities among normal breast tissue, breast tumors, spleen, or cecal contents during tumor progression (Figure 4a–d). However, unweighted UniFrac distance analysis showed clear segregation of micribiota between cecal content microbiota and other tissue-associated microbiomes (Figure 4e). Notably, the unweighted count showed at the 5-week time point, breast tumor and normal breast tissue microbiomes exhibited significantly higher beta diversity compared to cecal contents and spleen microbiomes, respectively (*p* < 0.01, Figure 4f).

These findings collectively indicate that tumor progression did not significantly alter alpha or beta diversity in normal breast tissue, breast tumors, or cecal contents, while spleen tissue demonstrated progressive increases in bacterial diversity.

### 3.3. The Abundance Changes in the Bacteria During Breast Tumor Growth

Our analysis identified a total of 50 bacterial phyla across all samples, with Proteobacteria, Firmicutes, Actinobacteria, and Bacteroidetes representing the dominant phyla in all four tissue types (Figure 5a). Longitudinal analysis revealed significant shifts in phylum-level abundances during tumor progression. In normal breast tissue, tumors, and cecal contents, we observed a progressive increase in Firmicutes abundance concomitant with decreasing Proteobacteria levels (*p* < 0.05). Conversely, spleen tissue exhibited an inverse pattern, with Proteobacteria increasing and Firmicutes decreasing over time (*p* < 0.05). At 3 weeks, Proteobacteria dominated breast tissues (70.1 ± 5.8%) and breast tumor tissue (73.3 ± 4.3%), while Firmicutes prevailed in spleen (72.8 ± 4.7%) and cecal contents (46.3 ± 5.4%). By 5 weeks, Actinobacteria and Bacteroidetes showed significant enrichment in breast tissues compared to spleen and cecal contents (*p* < 0.01). At 7 weeks, Firmicutes became predominant in tumors (63.4 ± 9.9%) and cecal contents (75.5 ± 0.8%), while Proteobacteria remained dominant in normal breasts (60.0 ± 9.9%) and the spleen (51.4 ± 4.0%) (Appendix A). Early dominance of *Sphingomonas* (3 weeks; 54.3 ± 13.0% and 51.7 ± 15.1%) shifted to *Bifidobacterium* (5 weeks; 10.5 ± 4.9% and 14.0 ± 8.4%) and later *Bacillus/Streptococcus* (7 weeks; 14.9% and 53.3%) in breast tissues and breast tumor tissues, respectively. *Sphingomonas* abundance decreased significantly in both normal and tumor tissues during progression (*p* < 0.01). The spleen showed progressive enrichment of potential pathogens, including *Proteus* and *Helicobacter* at 5 weeks (9.7 ± 4.6% and 2.9 ± 0.2%), and *Stenotrophomonas* at 7 weeks (15.0 ± 8.0%). There was a concurrent decrease in beneficial *Lactobacillus* (from 50.1% to 3.4%). Cecal content demonstrated increasing *Lactobacillus* (from 41.2% to 69.5%) with decreasing *Helicobacter* (from 13.5% to 2.2%) (Figure 5b, Appendix A).

Longitudinal species-level analysis (Figure 5c) revealed dynamically evolving microbial landscapes across tissue compartments during breast tumor progression. Breast tissue and tumors exhibited progressive diversification of enriched bacterial species, marked by the tumor-specific emergence of *Bacillus halodurans* at week 7, a signature completely absent in adjacent breast tissue. Concurrently, cecal contents demonstrated opposing trajectories for key species: *Helicobacter ganmani* abundance declined precipitously while *Lactobacillus reuteri* underwent substantial expansion, mirroring systemic microbial redistribution. Spleen tissue displayed phased colonization patterns, initially harboring *B. halodurans* at week 3, followed by week 5 enrichment of pathogenic bacteria, including *Proteus mirabilis*, *H. ganmani*, *Enterococcus faecalis*, and *Clostridium sporogenes*, with no subsequent dominant taxonomic shift observed through week 7.

Cross-tissue comparisons revealed striking ecological patterns. Microbial communities diverged distinctly across tissue types while maintaining shared foundational taxa, as evidenced by the spleen’s microbial composition encompassing nearly all genera enriched in breast, tumor, and gut ecosystems. Notably, Lactobacillus species, particularly *L. reuteri*, persisted ubiquitously across all anatomical sites, underscoring their potential role in maintaining microbial continuity during tumor-driven systemic alterations. These observations collectively delineate a spatiotemporal framework of microbial reorganization, where tissue-specific specialization coexists with cross-compartment taxonomic conservation during breast cancer progression.

### 3.4. The Differential Taxa Analysis During Breast Tumor Growth

LEfSe analysis (LDA score > 4, *p* < 0.05) delineated temporally resolved microbial signatures across four tissue compartments during breast tumor progression. In breast tumors, hierarchical taxonomic enrichment progressed from Proteobacteria phylum (LDA = 5.41, *p* = 0.04), genus Sphingomonas (LDA = 5.40, *p* = 0.01), family Sphingomonadaceae (LDA = 5.40, *p* = 0.01), order Sphingomomadales (LDA = 5.40, *p* = 0.01), and class Alphaproteobacterial (LDA = 5.39, *p* = 0.03) dominance at week 3 to Actinobacteria phylum (LDA = 4.94, *p* = 0.03) prominence at week 5, ultimately shifting to Bacilli-class (LDA = 5.46, *p* = 0.01), species *Bacillus halodurans* (LDA = 5.09, *p* = 0.04), species *Streptococcus azizii* (LDA = 5.04, *p* = 0.005), species *Bacillus litoralis* (LDA = 4.68, *p* = 0.005), and species *Paenibacillus humicus* (LDA = 4.17, *p* = 0.01) dominance at week 7 (Figure 6a). Normal breast tissue mirrored tumor-associated patterns at weeks 3–5 but diverged at week 7, retaining only *Streptococcus azizii* (LDA = 4.01, *p* = 0.02) and *Paenibacillus humicus* (LDA = 4.44, *p* = 0.04) as significantly enriched species (Figure 6b). Spleen tissue exhibited distinct phasic dynamics with Firmicutes phylum enrichment (LDA = 5.37, *p* = 0.03) at week 3, followed by transitional equilibrium at week 5 (Firmicutes 45.51%, Proteobacteria 45.90%) (Appendix A). By week 7, microbial reorganization emerged, featuring dual enrichment patterns—Proteobacteria (LDA = 5.24, *p* = 0.03), class Gamma-proteobacteria (LDA = 4.04, *p* = 0.01), order Cardiobacteriales (LDA = 4.27, *p* = 0.03), genus Ignatzschineria (LDA = 4.27, *p* = 0.03), Actinobacteria (LDA = 4.44, *p* = 0.04), and Corynebacteriales order (LDA = 4.05, *p* = 0.01)—coexisting with Sporosarcina genus (LDA = 4.30, *p* = 0.03) predominance (Figure 6c). Cecal content microbiota demonstrated progressive ecological succession (Figure 6d). Initial week 3 profiles demonstrated significant enrichment of Actinobacteria-derived lineages, specifically order Propionibacteriales (LDA = 4.11, *p* = 0.03) and family Nocardioidaceae (LDA = 4.13, *p* = 0.03). By week 5, microbial dominance shifted to Proteobacteria (LDA = 5.16, *p* = 0.02), class Delta-proteobacteria (LDA = 5.16, *p* = 0.02), and Clostridia (LDA = 4.53, *p* = 0.02) associated taxa, evidenced by enriched-order Desulfovibrionales (LDA = 5.16, *p* = 0.02), family Desulfovibrionaceae (LDA = 5.16, *p* = 0.02), and Clostridiales order (LDA = 4.54, *p* = 0.03), family Lachnospiraceae (LDA = 4.38, *p* = 0.01), genus *Clostridium* sp. (LDA = 4.22, *p* = 0.01). Week 7 marked Firmicutes (LDA = 5.27, *p* = 0.02) predominance through Lactobacillales order (LDA = 5.33, *p* = 0.02) expansion family Lactobacillaceae (LDA = 5.34, *p* = 0.02), and Lactobacillus genus (LDA = 5.34, *p* = 0.02). These temporal transitions establish distinct microbial ecologies across four tissues during breast oncogenesis.

Cladogram-based temporal profiling revealed progressive phylum-level microbial restructuring across four tissue compartments during breast tumor progression, demonstrating three-phase ecological succession. Initial analysis (week 3) identified Proteobacteria, Firmicutes, and Cyanobacteria as dominant phyla across all tissues (Figure 6e), transitioning by week 5 to expanded predominance of Bacteroidetes and Actinobacteria alongside persistent Proteobacteria/Firmicutes signatures (Figure 6f). Late-phase consolidation (week 7) established Actinobacteria–Proteobacteria–Firmicutes tripartite dominance (Figure 6g). Concomitantly, phylum-level diversification intensified temporally, with all tissue compartments demonstrating increased taxonomic complexity at advanced tumor stages, indicative of systemic microbial community evolution coordinated with oncogenic progression.

### 3.5. Functional Prediction of Microbiota During Breast Tumor Growth

Predictive functional profiling via the KEGG database (level 2) revealed temporally divergent metabolic trajectories across tissue compartments during breast tumor progression. In normal breast tissue, membrane transport pathways exhibited progressive amplification from weeks 3 to 7 (*p* < 0.05), whereas cell motility and signal transduction pathways declined transiently at week 5 (vs. weeks 3/7) (*p* < 0.05). Cofactor/vitamin metabolism pathways were significantly attenuated by week 7 compared to baseline (*p* < 0.05) (Figure 7a). Breast tumors demonstrated stage-specific alterations in which glycan biosynthesis/metabolism peaked at week 5, concurrent with suppression of lipid metabolism, alternative amino acid metabolism, and disease-related pathways such as infectious disease, cell growth/death, neurodegenerative diseases, and cancers (*p* < 0.05). Cofactor/vitamin metabolism was further diminished by week 7 (*p* < 0.05) (Figure 7b). Spleen tissue exhibited inverse dynamics where cell motility and signal transduction pathways surged at week 5 (*p* < 0.05) (vs. week 3), while membrane transport pathways declined by week 7 (*p* < 0.05). Late-phase reorganization featured enhanced lipid/amino acid metabolism at week 7 (vs. weeks 3/5) (*p* < 0.05), Neurodegenerative diseases and cancer pathways were enriched at week 7 (vs. week 3) (*p* < 0.05) (Figure 7c). Cecal contents showed progressive metabolic specialization where enrichment in replication/repair, nucleotide metabolism, alternative amino acid metabolism, enzyme families, and metabolic disease pathways contrasted with diminished cell motility (week 7 vs. week 3) (*p* < 0.05). Comparative week 7 vs. week 5 analysis revealed down-regulated carbohydrate metabolism, infectious disease, cell growth/death, and metabolic diseases pathways (*p* < 0.05), offset by enhancement in amino acid/energy metabolism and cell motility/signaling (*p* < 0.05) (Figure 7d). Collectively, these spatiotemporal profiles demonstrate conserved functional reprogramming across breast, tumor, and gut ecosystems, whereas splenic microbiota exhibited divergent pathway activation patterns, suggesting tissue-specific microbial metabolic adaptation during oncogenesis.

## 4. Discussion

Dysbiosis of microbiota in breast tissue and the gut has been implicated in breast cancer pathogenesis, with compositional variations linked to carcinogenesis and progression [4,6,36]. While prior studies predominantly focused on static comparisons between healthy and tumor tissues, our study uniquely delineates the spatiotemporal dynamics of microbiota across breast tumors, normal mammary tissue, the spleen, and the gut during tumor progression in a murine model. This longitudinal approach revealed critical insights into microbial translocation and tissue-specific adaptations, particularly in the spleen—a previously understudied compartment in breast cancer–microbiota interactions.

Notably, we observed stable alpha diversity in breast tumors, normal mammary tissue, and cecal contents during progression, contrasting with significant diversity increases in the spleen (*p* < 0.05). This finding diverges from previous reports of reduced or elevated diversity in breast tumors versus normal tissue [4,37], potentially attributable to our multi-timepoint analysis capturing dynamic equilibria. Cross-tissue comparisons at equivalent stages revealed higher microbial richness in breast-associated tissues than in spleen or cecal contents, consistent with Nejman et al. [29]. The microbiota of breast tumors included not only microbiota translocation from gut bacteria but also tumor–resident intracellular microbiota, which are integral components of the tumor tissue [17,21]. Breast cancer has a particularly rich and diverse microbiome which are present in both cancer and immune cells [29]. However, no significant differences emerged between normal and tumor tissues, highlighting the necessity of longitudinal profiling to resolve transient microbial states.

At the phylum level, temporal shifts mirrored tissue-specific ecological pressures. Breast tumors exhibited progressive Firmicutes enrichment and Proteobacteria depletion, while the spleen displayed inverse trends, a novel observation underscoring its distinct microbial regulation. The genera of *Lactobacillus*, *Streptococcus*, and *Staphylococcus* were principally observed in breast tumors, which had been proven to lead to chemotherapy and radiation resistance of tumors [20], promote neoplastic processes and metastasis of breast tumor [17,21], induce cell proliferation by increasing the biosynthesis of reactive oxygen species (ROS) and intracellular cholesterol [38,39]. The increased Bacteroides of gut microbiota elevated the risk of breast cancer recurrence [29], and *Bacteroides fragilis* enriched in breast tumors can secrete Bacteroides fragilis toxin to promote the invasion and metastasis of tumor cells in the breast duct through the pathway of β-catenin and the Notch1 axis [26]. Class Gamma-proteobacteria enriched in the spleen were capable of secreting cytolethal distending toxin (CDT) [39], which induces breaking of single-stranded DNA at low doses and double-stranded DNA at high doses, and further directly causes DNA damage and tumorigenesis [40]. The spleen’s unique microbiota, characterized by phased colonization of pathobionts (*Proteus, Helicobacter*) and declining beneficial Lactobacillus, correlated with splenomegaly and metastatic progression. These dynamics suggest microbial-driven microenvironmental dysregulation, potentially exacerbating systemic inflammation or immune dysfunction. Crucially, the spleen’s microbiota encompassed nearly all genera enriched in other tissues, including gut-associated Lactobacillus, Desulfovibrio, and Helicobacter, implicating gut-to-spleen translocation via compromised intestinal barriers [28,31]. The ubiquitous presence of *L. reuteri* across all tissues further supports systemic microbial continuity during oncogenesis.

Functional profiling revealed divergent metabolic trajectories. Breast tissues showed attenuated cofactor/vitamin metabolism, while splenic microbiota exhibited enhanced lipid/amino acid metabolism and cell motility pathways at late stages. These tissue-specific metabolic adaptations may differentially fuel tumor progression versus immune modulation. During tumor growth, the disease-related pathways such as infectious disease, cell growth/death, and metabolic diseases were reduced in tumor and gut microbiota and enhanced in splenic microbiota, which might promote immune and inflammatory response [33]. Cecal contents demonstrated progressive Lactobacillus enrichment alongside Helicobacter depletion, contrasting with splenic accumulation of gut-derived taxa—a finding aligning with the gut–breast axis hypothesis [41] and suggesting a potential gut–spleen axis warranting further investigation.

Our study advances prior work by identifying stage-specific microbial signatures that early Sphingomonas dominance in breast and breast tumor tissues transitions to Bacillus/Streptococcus in late-stage tumors, paralleled by spleen-specific enrichment of Stenotrophomonas and pathogenic Enterococcus. These temporal patterns, unreported in static analyses, highlight the importance of longitudinal sampling. Notably, Streptococcus and Lactobacillus enrichment in tumors aligns with their reported roles in chemotherapy resistance and metastasis [20,21], while splenic Gamma-proteobacteria might promote DNA damage through cytolethal distending toxins [42]—a mechanism potentially linking microbial dysbiosis to splenomegaly.

In conclusion, our multi-tissue, time-resolved analysis unveils dynamic microbial reorganization during breast cancer progression, with the spleen emerging as a critical hub for gut microbiota translocation and pathogenic colonization. These findings provide a framework for targeting tissue-specific microbial communities at defined disease stages, offering novel therapeutic avenues to modulate tumor microenvironments and systemic immune responses.

## 5. Conclusions

This study reveals spatiotemporal microbial dynamics across breast tumors, mammary glands, the spleen, and the gut in a murine breast cancer model. Key findings demonstrate tissue-specific microbial reorganization, with gut-associated genera (*Lactobacillus*, *Helicobacter*) translocating to mammary and splenic tissues, suggesting systemic microbial redistribution via compromised intestinal barriers. Temporal profiling identified stage-specific microbial shifts, including pathogenic colonization in the spleen linked to microenvironmental dysregulation. While the spleen exhibited unique microbial diversity patterns correlating with systemic complications, cross-tissue microbial crosstalk (*Sphingomonas* enrichment) highlighted interconnected microbial networks. These findings emphasized the role of microbial translocation in shaping tumor microenvironments and systemic disease manifestations. By mapping microbial trajectories, this study advanced understanding of microbiome-driven oncogenesis and underscored the potential of targeting gut-derived microbial pathways in favor of evaluating microbiota-based therapies to mitigate metastasis and systemic inflammation, offering novel strategies for breast cancer intervention. This study is limited to the spatiotemporal changes of the microbiota during breast tumor growth and metastasis. Further studies of how the bacteria in the gut, spleen, and tumor interact with the immune system by the technologies of metagenomics and single-cell sequencing will elucidate mechanisms of microbial–host interactions that influence breast cancer development and progression.

## Figures and Tables

**Figure 1 microorganisms-13-01632-f001:**
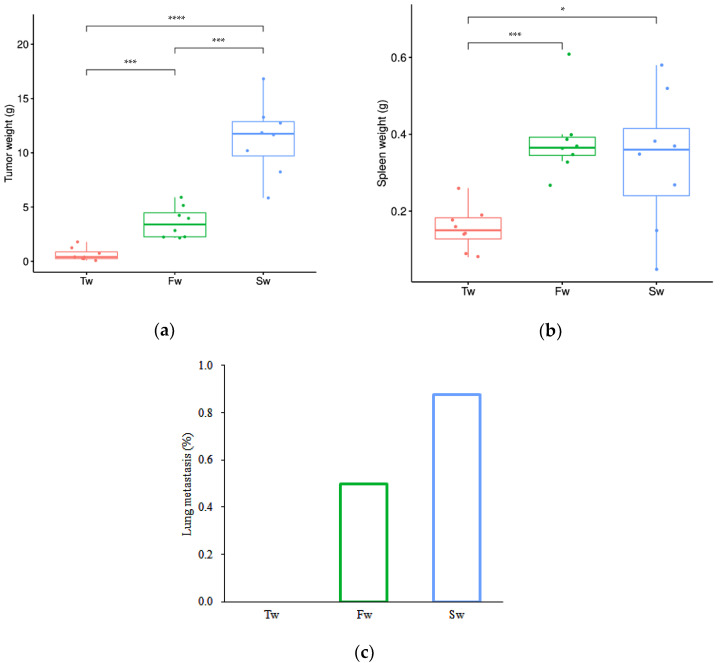
Breast tumor growth, splenomegaly, and lung metastasis at 3 weeks, 5 weeks, and 7 weeks. (**a**) Breast tumor growth curve. (**b**) Splenomegaly. (**c**) Lung metastasis rate. Data are presented as mean ± SD (*n* = 8). Statistical significance is tested using the Tukey test (* *p* < 0.05, *** *p* < 0.001, **** *p* < 0.0001). Note: Tw, Fw, and Sw represent 3 weeks, 5 weeks, and 7 weeks, respectively.

**Figure 2 microorganisms-13-01632-f002:**
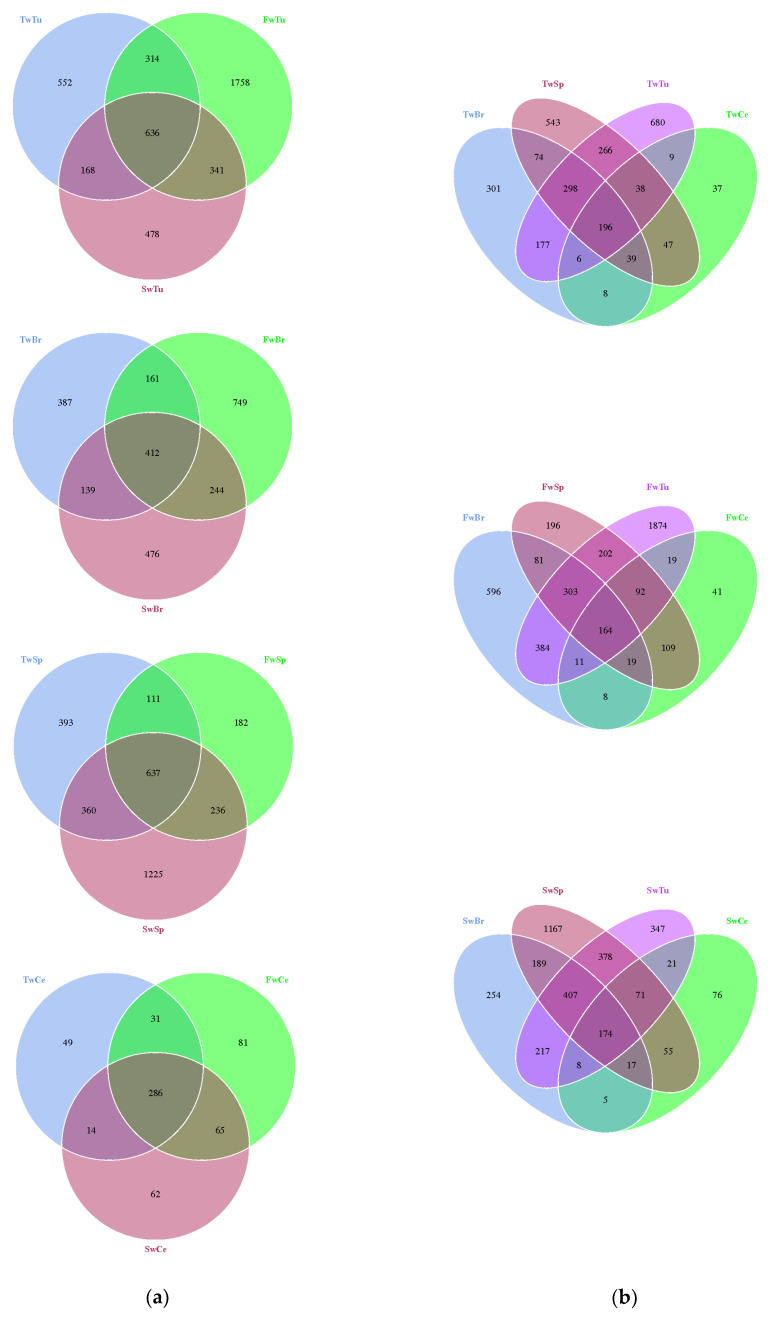
The distribution difference of the OTUs in breast tumor, normal breast tissue, spleen tissue, and cecal contents is shown by a Venn diagram. (**a**). Venn diagram of different tissues. (**b**). Venn diagram of three time points. Note: Tu, Br, Sp, and Ce represent breast tumor, normal breast tissue, spleen tissue, and cecal contents, respectively; Tw, Fw, and Sw represent 3 weeks, 5 weeks, and 7 weeks, respectively. TwTu, FwTu, and SwTu represent breast tumor at 3 weeks, 5 weeks, and 7 weeks, respectively; TwBr, FwBr, and SwBr represent normal breast tissue at 3 weeks, 5 weeks, and 7 weeks, respectively; TwSp, FwSp, and SwSp represent spleen tissue at 3 weeks, 5 weeks, and 7 weeks, respectively; TwCe, FwCe, and SwCe represent cecal contents at 3 weeks, 5 weeks, and 7 weeks, respectively. Same as below.

**Figure 3 microorganisms-13-01632-f003:**
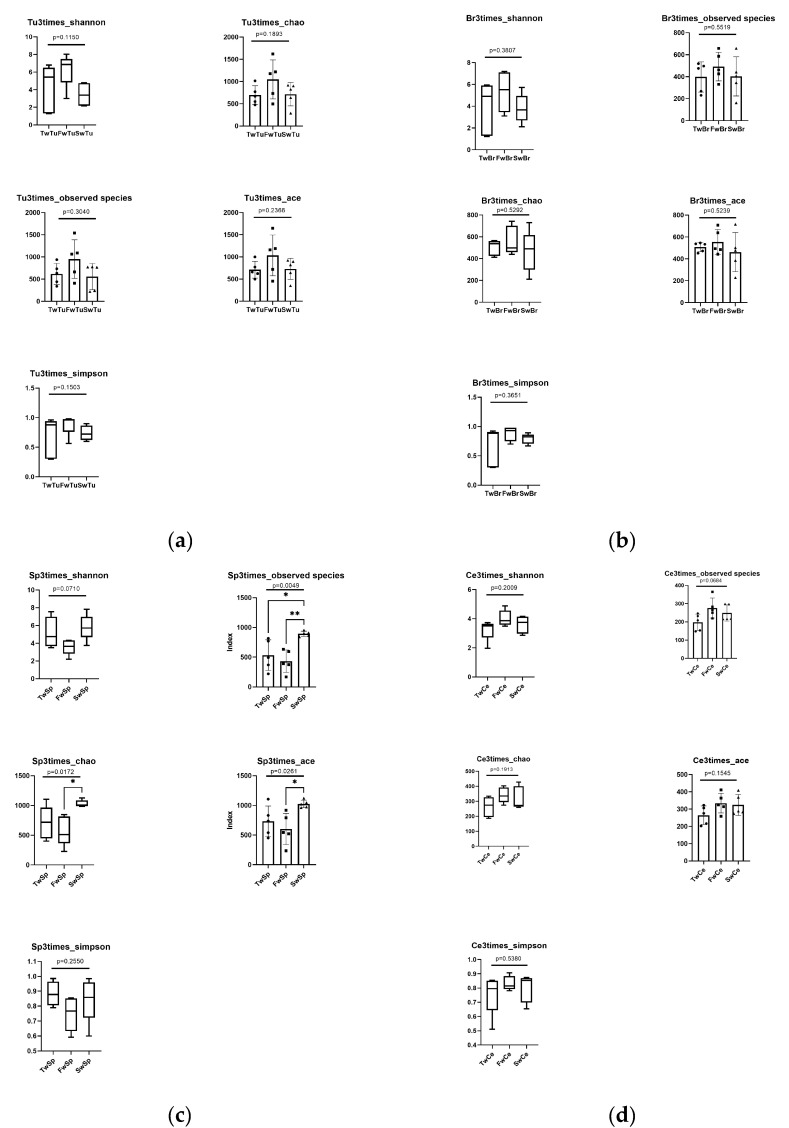
The analysis of alpha diversity at different stages and tissues. (**a**) Alpha diversity analysis of breast tumor microbiota. (**b**) Alpha diversity analysis of normal breast tissue microbiota. (**c**) Alpha diversity analysis of spleen tissue microbiota. Tukey test, * means *p* < 0.05, ** means *p* < 0.01. (**d**) Alpha diversity analysis of cecal contents microbiota. (**e**) Alpha diversity comparison of microbiota at the same stage across four tissues. Black dot, black square, and triangle represent the numerical values of alpha diversity indices at 3 weeks, 5 weeks, and 7 weeks respectively. Wilcox test, * means *p* < 0.05, ** means *p* < 0.01.

**Figure 4 microorganisms-13-01632-f004:**
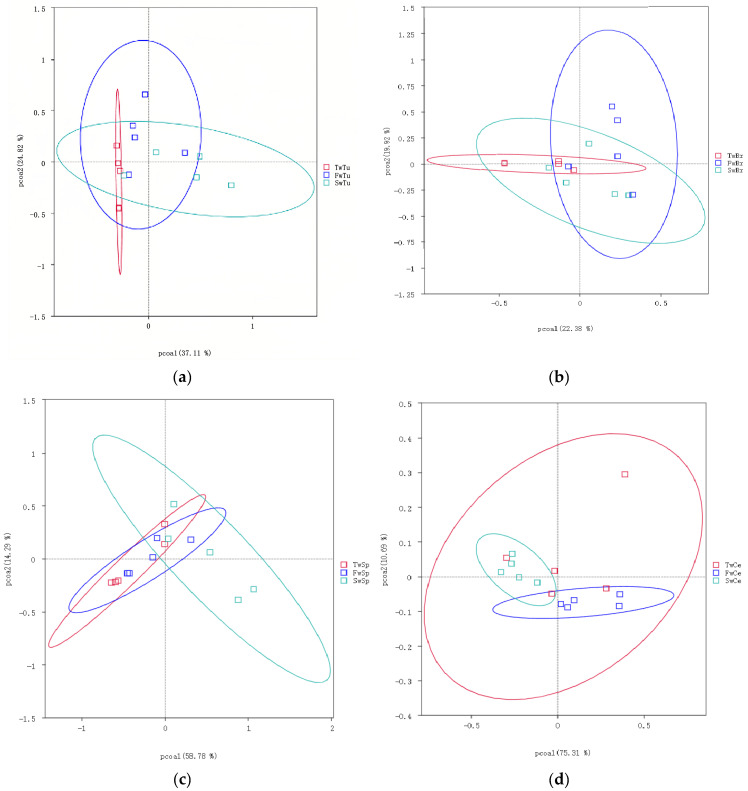
The analysis of beta diversity by principal coordinate analysis (PCoA) and unweighted unifrac distance at 3 weeks, 5 weeks, and 7 weeks. (**a**) Beta diversity analysis of breast tumor microbiota. (**b**) Beta diversity analysis of normal breast tissue microbiota. (**c**) Beta diversity analysis of spleen tissue microbiota. (**d**) Beta diversity analysis of cecal contents microbiota. (**e**) Beta diversity of microbiota among four tissues by unweighted unifrac distance. (**f**) Beta diversity statistics analysis of microbiota across four tissues with Tukey test; ** means *p* < 0.01.

**Figure 5 microorganisms-13-01632-f005:**
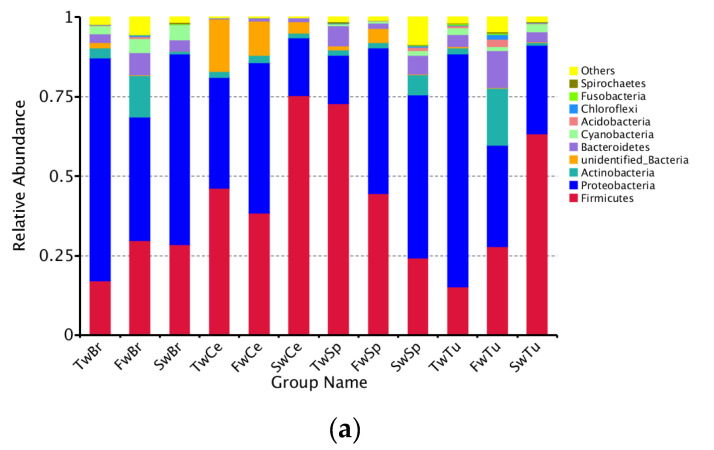
The abundance comparison of microbiota across normal breast tissue, breast tumor, spleen tissue, and cecal contents at the phylum, genus, and species level during tumor progression. (**a**) The microbial abundance comparison of microbiota at the phylum level in four tissues. (**b**) The microbial abundance comparison at the genus level in four tissues. (**c**) The microbial abundance comparison at the species level in four tissues.

**Figure 6 microorganisms-13-01632-f006:**
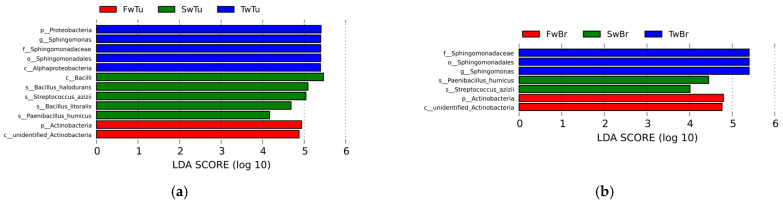
The differential microbial taxa in four tissues at different stages of breast tumor by LEfSe with the Tukey test (LDA score > 4, *p* < 0.05) and cladogram. (**a**) LDA value distribution in breast tumor at different stages. (**b**) LDA value distribution in normal breast tissue at different stages. (**c**) LDA value distribution in the spleen tissue at different stages. (**d**) LDA value distribution in cecal contents at different stages. (**e**) Cladogram in different tissues at 3 w. (**f**) Cladogram in different tissues at 5 w. (**g**) Cladogram in different tissues at 7 w.

**Figure 7 microorganisms-13-01632-f007:**
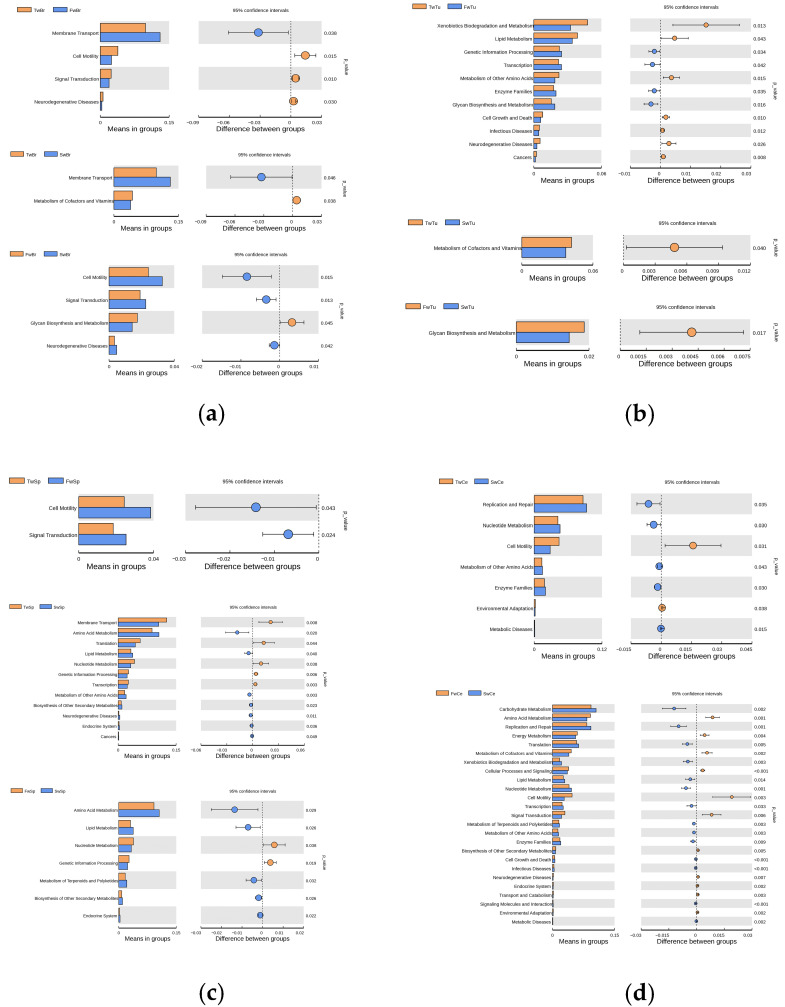
The analysis of the KEGG pathways at level 2 of microbiota at different stages of breast tumor. (**a**) KEGG pathways of normal breast tissue microbiota. (**b**) KEGG pathways of breast tumor microbiota. (**c**) KEGG pathways of spleen tissue microbiota. (**d**) KEGG pathways of cecal contents microbiota. Pillars in the right column indicate a significant increase, while those in the left column indicate a significant decrease in the abundances of the corresponding pathways; *t*-test, *p* < 0.05.

## Data Availability

The original contributions presented in this study are included in the article/Appendix A; further inquiries can be directed to the corresponding author.

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
