# Peer review of "Temporal and Spatial Dynamics of Tumor–Host Microbiota in Breast Cancer Progression"

_microorganisms, 2025, doi:10.3390/microorganisms13071632_

Round 1

Reviewer 1 Report

Comments and Suggestions for Authors

This interesting study addressed the relationship between microbiota dynamics during breast cancer progression using a murine model. However, it could be improved in several aspects:

There are missing descriptions and details of the animal model, techniques used with the animals and strain. Suggests to follow the ARRIVE 2.0 guidelines for the presentation of your data.

In the material and methods section it mentions "handling of rats" in the work presented, mice were used.

In the figures and figure captions it is not clear which groups are represented.

Author Response

Dear Reviewer:

   Thank you very much for taking the time to review this manuscript. We really appreciate your comments and suggestions to improve the quality of the manuscript. According to the suggestions, we have made corresponding modifications to our manuscript. All the points have been addressed and the manuscript has been revised appropriately. All the changes are highlighted in yellow throughout the revised manuscript. The point-by-point responses to the comments and suggestions are listed as below.

Comments 1: There are missing descriptions and details of the animal model, techniques used with the animals and strain. Suggests to follow the ARRIVE 2.0 guidelines for the presentation of your data.

Response 1: Thank you for pointing this out. We agree with this comment. Therefore, the descriptions and details of the tumor model have been added in the section of Materials and Methods as follow: Female mouse mammary tumor virus-polyoma middle tumor-antigen (MMTV-PyMT) transgenic mice (FVB/N-Tg(MMTV-PyVT)634Mul/J, 002374), which develop spontaneous breast tumor, were obtained from Shang Chai Lab of Westlake University. Twenty-four female Fvb mice, 6 weeks old, with an average initial weight of (25.3 ± 2.1) g were purchased from Experimental Animal Center of Zhejiang Province (Hangzhou, China). All animals were bred in Experimental Animal Center of Zhejiang University, housed in a specific pathogen-free conditions and given standard mouse chow. All animals were kept on a 12-hour light/12-hour dark cycle and at room temperature (Line 100-107, Page 3);  2.2. Murine model of breast tumor, A breast tumor from spontaneous tumor model MMTV-PyMT was screened. Dissociated spontaneous PyMT tumor cells were implanted into unilateral mammary gland of female Fvb mice at 1×105 cells/mouse, and the contralateral mammary gland was set as the control without implanting tumor cell. Tumor growth was monitored twice 1 week, and the mice was euthanized when tumors reach 2 cm in diameter as humane endpoints.(Line 114-120, page 3)

Comments 2: In the material and methods section it mentions "handling of rats" in the work presented, mice were used.

Response 2: Thank you for pointing this out. We have revised “2.1. Rats management” to “2.1. Mice management” in section of 2. Materials and Methods (Line 99, page 3).

Comments 3: In the figures and figure captions it is not clear which groups are represented.

Response 3: Thank you for pointing this out. We agree with this comment. Therefore we have illustrated the groups names in figure captions. Such as, Tw, Fw, Sw represent 3 weeks, 5 weeks and 7 weeks respectively in Figure 1(Line 197, page 5). Tu, Br, Sp, Ce represent breast tumor, normal breast tissue, spleen tissue and cecal contents respetively; Tw, Fw, Sw represent 3 weeks, 5 weeks and 7 weeks respectively; TwTu, FwTu, SwTu represent breast tumor at 3 weeks, 5 weeks and 7 weeks respectively; TwBr, FwBr, SwBr represent normal breast tissue at 3 weeks, 5 weeks and 7 weeks respectively; TwSp, FwSp, SwSp represent spleen tissue at 3 weeks, 5 weeks and 7 weeks respectively; TwCe, FwCe, SwCe represent cecal contents at 3 weeks, 5 weeks and 7 weeks respectively in Figure 2, The same as below Figure 3-7 (Line 258-262, page 6).

Reviewer 2 Report

Comments and Suggestions for Authors

This study demonstrates that the microbiota in mice with breast cancer changes over time in various tissues. While tumours and the gut saw an increase in Firmicutes, the spleen showed a rise in Proteobacteria and greater diversity, suggesting microbial translocation from the gut. The spleen serves as a reservoir for bacteria from other tissues, with distinct metabolic shifts occurring in response to immune responses. These findings highlight the role of microbiota in tumour progression and immune system changes.

Please carefully check for spelling errors and grammatical issues.

Abstract

 A clearer closing sentence could emphasize how these findings could guide microbiome-targeted intervention.

Introduction
Does not clearly explain what is unknown or how the study addresses issues that previous work does not.

The spleen is introduced too late and without a strong justification for its focus.

There is no explicit sentence summarizing the central hypothesis or the study's goals.

Methods

It’s unclear if mice were randomly assigned to groups or if blinding was applied during sample collection or analysis.

Although 24 mice were used, no explanation is provided for how the number was determined (e.g., power analysis).

What mouse strain was used? Were they monitored for clinical signs throughout the experimentation?

No mention of animal welfare measures, e.g., humane endpoints, housing conditions beyond temperature.

How many animals per group? It states 3, 5, and 7 weeks, but doesn't explain how many mice were sampled at each time point (e.g., whether n = 8 per time point was for all analyses).

There is no mention of how the tumour model was established, and no mention of control groups.

Are microbial profiles from individual animals shown, or were tissues pooled?

The kit used is for soil DNA (E.Z.N.A.™ Soil DNA Kit). This raises concerns about whether it's optimized for tissue samples, such as spleen and tumour, which may affect yield or bias.

 Illumina sequencing is mentioned, but platform (e.g., MiSeq, NovaSeq), read length, and depth per sample are not provided.

How were low-quality reads handled? Which software version of QIIME/UCLUST was used?

There is  no statement on rarefaction or normalization for alpha/beta diversity metrics.

It is not specified which datasets (e.g. alpha diversity, KEGG, LDA scores) were tested with t-tests or Tukey’s post hoc test.

Microbiome data often violate parametric assumptions; therefore, non-parametric tests, such as Kruskal–Wallis or PERMANOVA, should be considered.

No mention of statistical software or version information is provided.

The phrase “extreme significance” is non-standard. Scientific writing should use terms like “highly significant” or state the p-value.

Tumour weight and spleen weight are described as increasing, but no actual values (means ± SD).

There's no reference to values from non-tumour-bearing mice for comparison.

Values for the alpha diversity indices (e.g., Chao1, Shannon) are presented in the figures, not in the text.

There’s no confirmation that the data were rarefied or normalized before diversity calculation—a necessary step to avoid bias from uneven sequencing depths.

Statements like “Shannon and Simpson of breast tumour were significantly higher than those of spleen tissue…” are listed without interpretation.

While LDA score >4 is mentioned, actual LDA values and p-values for key taxa are not reported.

There’s no explanation of what the identified taxa imply for tumour biology, immune response, or gut-tissue axis.

P-values or fold changes for functional pathways are not provided, and it is unclear how significant or consistent these predicted shifts are.

The basis for the predictions is only loosely mentioned (KEGG Level 2 via PICRUSt), but no pipeline steps, confidence scores, or normalization approaches are described.

While functions are listed, there is no discussion of how these functional shifts relate to tumor biology, immune modulation, or metabolic crosstalk.

Conclusion

Briefly mention a limitation and call for complementary methods (e.g., metagenomics, immunophenotyping).

Need a specific concluding statement about how these findings may directly inform future diagnostics or treatments.

Comments on the Quality of English Language

Please carefully check for spelling errors and grammatical issues.

Author Response

Comments 1: Abstract, A clearer closing sentence could emphasize how these findings could guide microbiome-targeted intervention.

Response 1: Thank you for pointing this out. We agree with this comment. Therefore, a more lucid conclusion sentence has been substituted. The manuscript is revised in Abstract as follow: These results delineate the trajectory of microbiota translocation and colonization, demonstrate tissue-specific microbial redistribution during breast tumorigenesis, offering valuable implications for advancing microbiome-targeted cancer therapies.(Line 25-26,page 1).

Comments 2: Introduction,Does not clearly explain what is unknown or how the study addresses issues that previous work does not.

Response 2: Thank you for pointing this out. We agree with this comment. As far as “what is unknown or the study addresses issues that previous work does not”, it has been described in the text at Line 82-89, page 2. Additionally, we have updated “how this study addresses issues that previous work does not” as follow: In this study, we employed a murine breast cancer model to systematically investigate and map the temporal variations in microbial composition within breast tumors, normal mammary tissue, spleen, and cecal contents throughout tumor progression. These findings delineate the spatiotemporal distribution of bacteria at various stages of breast cancer to assess the role of microbial translocation in shaping tumor microenvironments and the mechanism that microbial–host interactions affected the  development and progression of breast cancer. ( Line 89-96, page 2) 

Comments 3: Introduction,The spleen is introduced too late and without a strong justification for its focus.

Response 3: Thank you for pointing this out. We agree with this comment. Therefore, we have cited the recent researches to elaborate on the importance of microbiota composition in spleen and the migration of gut bacteria to spleen tissue with breast tumor growth. The manuscript is revised in the section of Introduction as follow : A recent study showed the translocation of gut microbiota to the tissues of MLN and spleen  augmented extraintestinal anticancer immune responses [33], indicating spleen played a crucial role in tumor growth and therapy. (Line 78-81, page 2)  Simultaneously, the reference [33] was supplemented at Line 761-762. ( page 20)

Comments 4: Introduction,There is no explicit sentence summarizing the central hypothesis or the study's goals.

Response 4: Thank you for pointing this out. We agree with this comment. We have summarized this study’s goals as follow: In this study, we systematically investigate and map the temporal changes in microbial composition within breast tumors, normal mammary tissue, spleen, and cecal contents throughout tumor progression to assess the role of microbial translocation in shaping tumor microenvironments and the mechanism that the microbial–host interactions affected the development and progression of breast cancer. (Line 89-96, page 2

Comments 5: Methods,It’s unclear if mice were randomly assigned to groups or if blinding was applied during sample collection or analysis.

Response 5: In this study, mice was blinded at each time point during sample collection and analysis. (Line 121, page 3)

Comments 6: Although 24 mice were used, no explanation is provided for how the number was determined (e.g., power analysis).

Response 6: In this study, 24 female Fvb mice were used and 8 mice at each time point were used for sample collection and analysis (n=8). The sample size of mice per group (n=8) was referred to the references:

  1. [ChoiY.B.; Lichterman J.; Coughlin L.A.; et al. Immune checkpoint blockade induces gut microbiota translocation that augments extraintestinal antitumor immunity, Sci. Immunol. 2023, 8, eabo2003.]
  2. [Wang Y.;Wu Y.; Wang B.; et al. Bacillus amyloliquefaciens SC06 Protects Mice Against High-Fat Diet-Induced Obesity and Liver Injury via Regulating Host Metabolism and Gut Microbiota. Front. Microbiol. 2019,10, doi: 10.3389/fmicb.2019.01161]
  3. [Okubo R.;Nejman D.; Livyatan I.; et al. The human tumor microbiome is composed of tumor type-specific intracellular bacteria. Science. 2020, 368(6494), 973-980.]

   In these studies, the sample size per group were n=6-8 , n=6 of mice and n=5 of woman respectively for microbiota analysis in gut, spleen and tumor.

Comments 7: What mouse strain was used? Were they monitored for clinical signs throughout the experimentation?

Response 7: Female mouse mammary tumor virus-polyoma middle tumor-antigen (MMTV-PyMT) transgenic mice (FVB/N-Tg(MMTV-PyVT)634Mul/J, 002374) and female Fvb mice were used in this study. (Line100-103, page 3). We monitored the animals throughout the experimentation for clinical signs such as tumor diameter, feed intake, weight, capacity of action.

Comments 8: No mention of animal welfare measures, e.g., humane endpoints, housing conditions beyond temperature.

Response 8: Following the suggestion. We have mentioned the animal welfare measures in the section of Materials and Methods, The manuscript is revised as follow: All animals were bred in Experimental Animal Center of Zhejiang University, housed in a specific pathogen-free conditions and given standard mouse chow. All animals were kept on a 12-hour light/12-hour dark cycle and at room temperature.(Line 105-108, page 3). The mice were euthanized when tumors reach 2 cm in diameter as humane endpoints. (Line 119-120, page 3)

Comments 9: How many animals per group? It states 3, 5, and 7 weeks, but doesn't explain how many mice were sampled at each time point (e.g., whether n = 8 per time point was for all analyses).

Response 9: In this study, blinding was applied during sample collection at each time point.   Tissue samples of breast tumor and spleen isolated from 8 female Fvb mice were aseptically collected and weighed for tumor growth and splenomegaly analysis at each time point (n=8 ). (Line 121-123, page 3). Tissue samples of the breast tumor tissue, contralateral healthy breast tissue, spleen tissue and cecal contents) of five female Fvb mice were isolated for microbiome analysis (n=5) at each time point. (Line 129-130, page 3).

Comments 10: There is no mention of how the tumour model was established, and no mention of control groups.

Response 10: Thank you for pointing this out. We missed to describe the imformation of the tumor model in the manuscript. Therefore, we have provided the descriptions and details of the tumor model in the section of Materials and Methods. The manuscript is revised as follow: 2.2. Murine model of breast tumor. A breast tumor from spontaneous tumor model MMTV-PyMT was screened. Dissociated spontaneous PyMT tumor cells were implanted into unilateral mammary gland of female Fvb mice at 1×105 cells/mouse, and the contralateral health mammary gland from the same individual without implanting tumor cell was set as the control. Tumor growth in mice was monitored twice 1 week, and the mice was euthanized when the diameter of tumors reach 2 cm as humane endpoints. (Line 114-120. page 3)

Comments 11: Are microbial profiles from individual animals shown, or were tissues pooled?

Response 11: In this study, microbial profiles are shown from individual mice, tissues were not pooled.

Comments 12: The kit used is for soil DNA (E.Z.N.A.™ Soil DNA Kit). This raises concerns about whether it's optimized for tissue samples, such as spleen and tumour, which may affect yield or bias.

Response 12: Thank you for pointing this out. We agree with this comment. Therefore, we re-checked and found due to a typing error, "E.Z.N.A.® Bacterial DNA Kit" was mistakenly written as "E.Z.N.A.® soil DNA Kit". After the author's verification and confirmation, the DNA kit used should be the E.Z.N.A.® Bacterial DNA Kit rather than the E.Z.N.A.® soil DNA Kit. We have modified the E.Z.N.A.TM Soil DNA Kit to E.Z.N.A.® Bacteria DNA Kit  (Line 133, page 3).

Comments 13: Illumina sequencing is mentioned, but platform (e.g., MiSeq, NovaSeq), read length, and depth per sample are not provided.

Response 13: Following the suggestion, the information of sequencing platform, read length, and depth per sample have been supplemented in the manuscript as follow: In this study, the V3-V4 region of the 16S rRNA gene was amplified by PCR and sequenced by Illumina high-throughput sequencing on the IonS5TMXL flatform (Line 144, page 3). The high microbial community coverage (Goods-coverage range from 0.986 to 0.997) indicated the eligible sequencing quality across all samples (Figure S1).Average read lenght was 409 bp.(Line 201-203, page 5).

Comments 14: How were low-quality reads handled? Which software version of QIIME/UCLUST was used?

Response 14: Raw FASTQ file reads were quality-filtered using Cutadapt (V1.9.1) to remove low quality part reads and chimeras to get the Clean Reads. (Line 147-148, page 6). The Version 1.9.1 of QIIME software was used in data analysis. ( Line 158, page 6).

Comments 15: There is no statement on rarefaction or normalization for alpha/beta diversity metrics.

Response 15: Thank you for pointing this out , and the rarefaction for diversity metrics have been stated in the manuscript as follow: The Alpha diversity indices including Observed- species, Chao1, Shannon, Simpson, ace, and Goods-coverage,were calculated using Qiime software (Version 1.9.1) . The rarefaction curves were plotted using R software (Version 2.15.3). (Line 153- 160, page 4). Additionally, the rarefaction curves of all samples have provided as supplementary file: The rarefaction curves of all samples indicated the sequencing data volume were reasonable (Figure S2).( Line 266-267, Line 687)

Comments 16: It is not specified which datasets (e.g. alpha diversity, KEGG, LDA scores) were tested with t-tests or Tukey’s post hoc test.

Response 16: Thank you for pionting this out. Methods of statistical tests to the data have been defined in figure legends. Therein, the data of alpha and beta diversity were analyzed with Tukey’s test and wilcox test, ,those of LDA scores, relative abundance, breast tumor growth and splenomegaly were analyzed with Tukey’s test, and that of KEGG was tested with t-tests.

Comments 17: Microbiome data often violate parametric assumptions; therefore, non-parametric tests, such as Kruskal–Wallis or PERMANOVA, should be considered.

Response 17: Thank you for your comment. Methods of non-parametric test to the data have been added in the manuscript as follow: The difference analysis of the Alpha diversity index and beta diversity conducted parametric tests with Tukey- test and non-parametric tests with wilcox test respectively. (Line 161-163,page 4). Additionally, we have replaced Tukey test result with Wilcox test result in the Figure 3e, synchronously the original images of Figure 3e have been re-uploaded(Line309-339,page8), the results in the text were modified.( Line 275-281, page 8)

Comments 18: No mention of statistical software or version information is provided.

Response 18: Thank you for your comment. The software of SPSS 25.0 was used for data analysis. The manuscript has been revised as follow: The data statistics was performed with SPSS 25.0 software (SPSS Inc., Chicago, IL, United States). ( Line 175-176, page 4)

Comments 19: The phrase “extreme significance” is non-standard. Scientific writing should use terms like “highly significant” or state the p-value.

Response 19: Thank you for pointing this out. We have modified “extreme significance” to “highly significant” and concurrently state the p-value throughout the text.(Line 175, 277, 281)

Comments 20: Tumor weight and spleen weight are described as increasing, but no actual values (means ± SD).

Response 20: According to the suggestion, we have added the actual values (means ± SD) of tumor weight and spleen weight in the section of Result (Line 182-184, page 4)

Comments 21: There's no reference to values from non-tumour-bearing mice for comparison.

Response 21: Thank you for pointing this out. In this study tumor cell implanted into unilateral mammary gland of female Fvb mice, the contralateral healthy mammary gland from the same individual was set as the control without implanting tumor cell. (Line 117-118, page 3). Therefore, there is no reference to values from non-tumour-bearing mice for comparison.

Comments 22: Values for the alpha diversity indices (e.g., Chao1, Shannon) are presented in the figures, not in the text.

Response 22:  Following the suggestion, Values for the alpha diversity indices have benn presented in the text. (Line 271-282, page 7)

Comments 23: There’s no confirmation that the data were rarefied or normalized before diversity calculation—a necessary step to avoid bias from uneven sequencing depths.

Response 23: Thank you for pointing this out. We have clarified the normalized step before diversity calculation. The manuscript is revised as follow: Normalized processing was carried out based on the standard of the least amount of data in the sample. The alpha/beta diversity analysis were conducted based on the normalized OTU. (Line 153-155,page 4)

Comments 24: Statements like “Shannon and Simpson of breast tumour were significantly higher than those of spleen tissue…” are listed without  interpretation.

Response 24: According to the suggestion, we have added the interpretation in the section of Discussion. The manuscript is revised as follow: The microbiota of breast tumor included not only microbiota translocation from gut bacteria and but also intracellular bacteriatumor-resident intracellular microbiota which are integral components of the tumor tissue [17,21]. Breast cancer has a particularly rich and diverse microbiome which are present in both cancer and immune cells [31]. (Line 608-612, page 16 )

Comments 25: While LDA score >4 is mentioned, actual LDA values and p-values for key taxa are not reported.

Response 25: According to the suggestion, we have supplemented the actual LDA values and p-values for key taxa in the section of Results (Line 458-482, page 13)

Comments 26: There’s no explanation of what the identified taxa imply for tumour biology, immune response, or gut-tissue axis.

Response 26: Thank you for pointing this out. We have explained the relationship between the identified taxa and tumor development, immune response, or gut-tumor axis in the section of Discussion. The manuscript is revised as follow: The genera of Lactobacillus, Streptococcus, and Staphylococcus were principally observed in breast tumor which had been proven to lead  chemotherapy and radiation resistance of tumors [20], promote neoplastic processes and metastasis of breast tumor [17,21], induce cell proliferation by increasing the biosynthesis of reactive oxygen species (ROS) and intracellular cholesterol [38,39]. The increased Bacteroides of gut microbiota elevated the risk of breast cancer recurrence [31] and Bacteroides fragilis enriched in breast tumor can secrete Bacteroides fragilis toxin to promote the invasion and metastasis of tumor cell in breast duct through the pathway of β-catenin and Notch1 axis [26]. Class Gamma-proteobacteria enriched in spleen, was capable of secreting cytolethal distending toxin (CDT) [39] which induce breaking of single-stranded DNA at low doses and double-stranded DNA at high doses, and further directly cause DNA damage and tumorigenesis [40]. (Line 618-629, page 17). Simultaneously, the references [38-40] was supplemented. (Line 770-774, page 20) 

Comments 27: P-values or fold changes for functional pathways are not provided, and it is unclear how significant or consistent these predicted shifts are.

Response 27: According the suggestion, we have provided the P-values for functional pathways (Line 530-549, page 16), and emphasized the shifts of the disease-related pathways. (Line 535-537, 542,547-548,page 16)

Comments 28: The basis for the predictions is only loosely mentioned (KEGG Level 2 via PICRUSt), but no pipeline steps, confidence scores, or normalization approaches are described.

Response 28: Thank you for pointing this out. We have described steps as follow: Microbial functions were predicted by PICRUSt (version 1.1.0), basing on the OTU tree and OTU genetic information in the Greengene database. Predicted functional pathways were annotated by using KEGG at level 2. (Line 167-170, page 4)

Comments 29: While functions are listed, there is no discussion of how these functional shifts relate to tumor biology, immune modulation, or metabolic crosstalk.

Response 29: Thank you for pointing this out, We have improved the discussion of functions with tumor biology, immune modulation. The manuscript is revised as follow: During tumor growth, the disease-related pathways such as infectious disease, cell growth/death and metabolic diseases reduced in tumor and gut microbiota, and enhanced in splenic microbiota which might promote immune and inflammatory response [33]. (Line 643-646, page 17)

Comments 30: Conclusion, Briefly mention a limitation and call for complementary methods (e.g., metagenomics, immunophenotyping).

Response 30: According the suggestion, we have mentioned a limitation and future work in depth analysis. The manuscript is revised as follow: This study is limited to the spatiotemporal changes of the microbiota during breast tumor growth and metastasis. Further studies of how the bacteria in the gut ,spleen, and tumor interact with the immune system by the technologies of metagenomics and single cell sequencing will elucidate mechanisms of microbial-host interactions influence breast cancer development and progression.(Line 681-685, page 18

Comments 31: Need a specific concluding statement about how these findings may directly inform future diagnostics or treatments.

Response 31: Following the suggestion, we have provided a concluding statement. The manuscript is revised as follow: These findings emphasized the role of microbial translocation in shaping tumor microenvironments and systemic disease manifestations. By mapping microbial trajectories, this study advanced understanding of microbiome-driven oncogenesis and underscored the potential of targeting gut-derived microbial pathways in favour of evaluating microbiota-based therapies to mitigate metastasis and systemic inflammation, offering novel strategies for breast cancer intervention. (Line 675-681,page 18)

Point 1: Please carefully check for spelling errors and grammatical issues

Response 1: Thank you for pointing this out. We have re-checked the whole manuscript carefully and corrected the errors in grammar and spelling which were in red throughout the revised manuscript. Part of the errors corrected were listed below:

Line 15-16: Add “the” before spleen, and “a” before “decline”

Line 32: Add “ is” before “the second dominant factor ”

Line 130: “Isolated of five mice ” is modified to “isolated from five mice”

Line 397,449: “cross” is modified to “across”

Round 2

Reviewer 2 Report

Comments and Suggestions for Authors

Accept for publication